# The Interactions between COVID-19 Cases in the USA, the VIX Index and Major Stock Markets

Simon Grima [1,*] , Letife Özdemir [2] , Ercan Özen [3] and Inna Romānova [4]

1 Department of Insurance, Faculty of Economics, Management and Accountancy, University of Malta, MSD 2080 Msida, Malta

2 Department of Logistics Management, School of Applied Sciences, University of Afyonkocatepe, Afyonkarahisar 03200, Turkey; letifeozdemir@aku.edu.tr

3 Department of Finance and Banking, Faculty of Applied Sciences, University of Usak, Uşak 64200, Turkey; ercan.ozen@usak.edu.tr

4 Faculty of Business, Management and Economics, University of Latvia, LV-1586 Riga, Latvia; inna.romanova@lu.lv

* Correspondence: simon.grima@um.edu.mt

**Abstract:** With this study, we aimed to determine (1) the effect of the daily new cases and deaths due to the COVID-19 pandemic in the United States on the CBOE volatility index (VIX index) and (2) the effect of the VIX index on the major stock markets during the early stage of the pandemic period. To do this, we collected and analysed the daily new cases and death numbers during the COVID-19 pandemic period in the United States and the country indexes of the USA (DJI), Germany (DAX), France (CAC40), England (FTSE100), Italy (MIB), China (SSEC) and Japan (Nikkei225) to determine the impact of the VIX index on the major stock markets. We then subjected this data to the Johansen co-integration test and the fully modified least-squares (FMOLS) method. The results indicated that there was co-integration between the VIX and the COVID-19 pandemic and that there was co-integration between the VIX index and major indexes, except for the CAC 40 and MIB. Moreover, the results showed that the new COVID-19 cases in the USA had a higher impact on the VIX than cases of deaths during the same period.

**Keywords:** COVID-19 pandemic; fear index; co-integration test; fully modified least squares method; stock markets

## 1. Introduction

In December 2019, the coronavirus disease 2019 (COVID-19) outbreak began in the city of Wuhan, Hubei region, China. As of 20 March 2020, the virus had already affected more than 500,000 people in more than 60 countries, with around 16% deaths due to this virus. Around 5% of the infected patients were in a critical or serious condition, while there seemed to be a recovery rate of 84% (Worldometer 2020).

On 3 February 2020, the Shanghai stock market plunged 8% following the general distress over COVID-19 in China. This shocking disruption rapidly spread to international financial markets. For example, the United States (U.S.) stock prices logged their lowest level in February and the S&P 500 plummeted 4.4% on 28 February 2020. Initially ignored by many countries, the COVID-19 effect was raising serious concerns due to its rapid propagation outside China (Albulescu 2020a, 2020b).

Can this be the "worst financial crisis the world has ever seen since 1929?" This is what some analysts, such as Elliot (2020), believe. He noted in his article that Stephen Isaccs (2020) of Alvine Capital highlighted that COVID-19 is "unprecedented", with record levels of leverage and overbought stocks. Moreover, he highlights that both Goldman Sachs and

HIS Markit revised their forecasts downwards for the world's real GDP growth in 2020 to 1.25% and 0.7% respectively[1].

Herron and Hajric (2020) highlighted the panic mode in the markets and noted that stock markets plunged 12% amid COVID-19 fears. However, Desjardins (2020) showed that some markets (of holdings, such as Zoom Video Communications (ZM), Domino Pizza (DPZ), Campbell Soup Company (CPB), Teladoc Health, Inc. (TDOC), The Clorox Company (CLX), Everbridge, Inc. (EVBG) and Virtu Financial, Inc. (VIRT)) were thriving from this situation by creating a vortex to suck up the alternate market universe, which he called "the pandemic economy". He said that, on average, these companies saw an upward bump of 12.7%.

As we note, during a pandemic period, such as the COVID-19 pandemic, the factors affecting portfolios change; therefore, to achieve a well-designed portfolio, we need to understand the impact of cases and deaths on market risk and the stock markets.

## 2. Literature Review

Most studies carried out by authors, such as Haacker (2004), Lee and McKibbin (2004), Loh (2006), Kauffman and Weerapana (2006), McKibbin and Fernando (2020), Fernandes (2020), Albulescu (2020a), Albulescu (2020b), Ramelli and Wagner (2020b), Zeren and HIZARCI (2020), on pandemics and their effect on the economies and financial markets relate mainly to HIV/AIDS and SARS. However, there is a small but growing literature emerging on the impact of COVID-19 on the stock markets.

In his research, Fernandes (2020) highlights that this pandemic (COVID-19) differs from other global crises since it is a global pandemic, which does not focus only on the low-to-middle-income countries, the interest rates are at historical lows, the world is much more integrated than before and there are spillover effects throughout the supply chains that disrupt the demand and supply.

Haacker (2004) showed that HIV/AIDS affected economic units, such as businesses, households, governments, labour supply decisions, labour efficiency and household income. He noted that business costs, public expenditure on healthcare and support of disabled and children orphaned by AIDS increased, causing budget deficits in certain countries. Kauffman and Weerapana (2006) revealed that bad news about HIV/AIDS in the Republic of South Africa had a negative effect on the value of the South African rand against the U.S. dollar.

The SARS epidemic had significant effects on various economies that were caused by the reductions in total demand for various goods and services, increases in business operating costs and increases in each country's risks, which in turn increased the risk premiums. Although the number of infected persons and deaths due to this epidemic was not the same in all countries, the impact on a global scale, which resulted in a cost of 54 billion USD in 2003, was significant (Lee and McKibbin 2004). The capital outflow in Hong Kong and China rose to 1.4% and 0.8% of GDP, respectively, and their risk premium was increased by 200 basis points (Lee and McKibbin 2004).

On the other hand, Loh (2006) showed that the SARS pandemic increased airline stocks' volatility with lower mean returns in certain countries. However, it had a negligible impact on the mean stock market returns and had no significant long-run implications. The author highlighted that airline stocks tended to take on an "aggressive" characteristic in the presence of any pandemic outbreak.

McKibbin and Fernando (2020) used the G-Cubed multi-country model to calculate the effect of COVID-19 on the global economy in the early stages of the pandemic while the outbreak was still only in China. They explored seven different scenarios of how COVID-19 might develop and influence macroeconomic outcomes and financial markets using global hybrid dynamic stochastic general equilibrium (DSGE) and computable general equilibrium (CGE) models developed by McKibbin and Wilcoxen (1999, 2013).

---

[1]    Growth below 2% is classified as a global recession.

They found that the impact of the spread of COVID-19 on the financial risk in the United States was high relative to the G-20 and OECD countries and that England and several developing countries, such as Argentina, South Africa and Turkey, had a higher degree of relative financial risk resulting from the spreading of COVID-19 than the United States. Therefore, they note that the spreading of COVID-19 to these countries would deeply affect the financial markets. Using these scenarios, they demonstrated that even a controlled outbreak could have a significant impact on the global economy in the short run and that the costs may be avoided by greater investment in public health systems.

Fernandes (2020) analysed the economic effects of the COVID-19 outbreak on the world economy as of 22 March 2020. The author noted that most stock markets on that date collapsed and registered their largest recorded one-day falls on record; some well-known companies saw their stock prices fall by more than 80% in a few days. He noted that in the United States, stock markets saw their worst performance with a fall of over 25% and British markets had the largest hit of all developed markets with a decline of more than 35%. He further highlighted that the impact of the outbreak of this pandemic (COVID-19) was being underestimated and could not be compared to other pandemics, such as SARS and the 2008/2009 financial crisis. His findings demonstrated that in a mild scenario, GDP would take a hit ranging from 3–5% depending on the country and that specifically service-oriented and tourism-reliant economies would be negatively affected, with the largest job losses.

In the first 40 days of international monitoring of the COVID-19 outbreak, Albulescu (2020a) found in his study that the death ratio positively influenced the VIX and that this influencing effect was stronger outside China. In another study, Albulescu (2020b) found that following the outbreak of the virus infections resulted in a marginally negative impact on crude oil prices in the long run. Moreover, COVID-19 also has an indirect effect on crude oil prices when the volatility of the financial markets was amplified.

Ramelli and Wagner (2020b) noted that after the outbreak of COVID-19 in China and the United States, industry stock returns faced disruptions. Stock returns in some sectors, such as telecom services, healthcare and software services, were on the increase in China and the United States. However, stock returns in other sectors, such as energy, transportation, insurance, real estate, retailing and automobiles, were on the decrease. Ramelli and Wagner (2020a, 2020b) stated that the Chinese and the U.S. stock markets were quick to respond to concerns about the possible economic consequences of the COVID-19 pandemic and explain that this resulted because the investors became increasingly worried about corporate debt and liquidity, which mutated into an economic crisis that was augmented through financial channels.

Zeren and HIZARCI (2020) investigated the co-integration relationship between COVID-19 cases and some selected stock markets. They used data between 23 January 2020 and 13 March 2020. Their findings showed that there was a co-integration relationship between the COVID-19 cases and the SSE, KOSPI and IBEX35, and no relationship with the FTSE MIB, CAC40 or DAX30 was found. This result showed the geographical effect of COVID-19 on stock markets as the virus spread to the European countries at the beginning of March.

Cheng (2020) highlighted that stock investors underpriced the risk of the COVID-19 pandemic and showed that trading in the VIX futures market was a step behind in relating the risks. He noted that although the cases of COVID-19 were increasing rapidly in Europe starting in March and there were reports of community spreading and deaths in the United States, the S&P 500 had declined slightly, while the price of the VIX had risen by 42%.

Onali (2020) investigated the impact of the COVID-19 cases and deaths on the U.S. stock market, specifically the Dow Jones and the S&P500 indices. He allowed for changes in trading volume and volatility expectations, as well as day-of-the-week effects. Using a GARCH(1,1) model on the data collected during the period from 8 to 9 April 2020, he found that, except for China (where reported cases had an effect), changes in the number of cases and deaths in the United States and six other countries strongly affected by the

COVID-19 crisis did not have an impact on the U.S. stock market returns. However, he further noted that his findings evidenced a positive impact for some countries on the conditional heteroscedasticity of the Dow Jones and S&P500 returns, that the VAR models indicated that reported deaths in Italy and France had a negative impact on stock market returns, a positive impact on the VIX returns and that the magnitude of the negative impact of the VIX on stock market returns spiked threefold.

In their study, Baker et al. (2020) used textual analysis to quantify the impact of news on COVID-19 cases on the volatility in the stock market, specifically the Dow Jones index, using data up to 9 April 2020 for cases and deaths in the United States, China, France, Iran, Italy, Spain and the United Kingdom. They found that the volatility impact was much larger during this pandemic than during similar disease outbreaks and that the COVID-19 crisis had caused a change in the relationship between volatility expectations and stock market returns.

In a further study, Yilmazkuday (2020) investigated the impact of the number of deaths related to COVID-19 on the S&P500 index. He used daily data collected during the period between 31 December 2019 and 1 May 2020 and used a structural vector autoregression model, using a measure for the global economic activity, the spread between 10-year treasury maturity and the federal funds rate. The results showed that a 1% increase in the U.S. cumulative daily COVID-19 cases resulted in around a 0.01% cumulative reduction in the S&P 500 Index after one day and around a 0.03% reduction after one month, with the largest observations during March 2020.

Using data up to June 2020 from the aggregated stock market and the dividend futures, Gormsen and Koijen (2020a, 2020b) measured the investors' expectations on economic growth as a response to the COVID-19 outbreak and the subsequent policy responses until June 2020. They showed that the growth expectations across maturities evolved and provided a simple model for understanding the joint dynamics of short-term dividend futures, stock markets and bond markets.

As noted from the literature, there are a limited number of papers that study the impact of COVID-19 cases on the VIX index and the impact of the VIX index on the major stock markets during the COVID-19 pandemic, especially using U.S. data.

### 3. Aim and Data

Therefore, given the above discussion and the uncertainties during this COVID-19 "pandemic economy", we aimed to answer the following research questions: (1) What was the impact of COVID-19 data on the VIX index in the United States?

We hypothesised that:

**Hypothesis 1 (H1).** *Bad news on COVID cases and deaths in the United States did not influence the VIX index.*

**Hypothesis 2 (H2).** *Bad news on COVID cases and deaths in the United States influenced the VIX index.*

(2) What was the impact of the VIX on the major world stock exchanges during the same period?

We hypothesised that:

**Hypothesis 3 (H3).** *As the VIX index increased, the major world stock exchange prices would also increase.*

**Hypothesis 4 (H4).** *As the VIX index increased, the major world stock exchange prices would decrease.*

To address the first research question, we used daily new case and death numbers during the COVID-19 pandemic in the United States. Since the cases started to increase as

of 27 January 2020 in the USA, we collected data for the period of 27 January 2020 to 29 May 2020 (the analysis period).

Then, to determine the VIX effect on the major stock exchanges during the COVID-19 pandemic period, we collected and analysed daily closing price data for the USA (DJIA), Germany (DAX), France (CAC40), England (FTSE100), China (SSEC), Japan (Nikkei225) and Italy (MIB) for the period between 2 January 2020 and 29 May 2020. The COVID-19 data was collected from www.worldometers.info (accessed on 13 May 2021) (Worldometer 2020), while the index data was collected from www.investing.com (accessed on 13 May 2021) (Investing 2020). We conducted the analysis presented below to determine the effect of the COVID-19 pandemic on the VIX index and the effect of the VIX index on the major stock markets during the pandemic period.

This study is especially equally important for those portfolio and fund managers who build their portfolios around the market indexes and academics who study the effects of specific announcements. In addition, this can be of benefit to risk managers, underwriters and actuaries who might need to revise their measurements in line with new data and information and for portfolio diversification and portfolio management decisions. The research period was chosen to eliminate as much as possible any noise that may have been introduced due to pandemic fatigue and the news of vaccines and antiviral medicines since there was no fatigue yet and no news about the vaccines in this period; the concentration of everyone was on the element of uncertainty. We wanted to specifically understand how news of the widespread pandemic affected the so-called fear index and major markets.

## 4. Methodology and Empirical Results

### 4.1. Effect of the COVID-19 Pandemic on the VIX Index

The VIX index and daily new case and death numbers during the COVID-19 pandemic are provided in Figure 1.

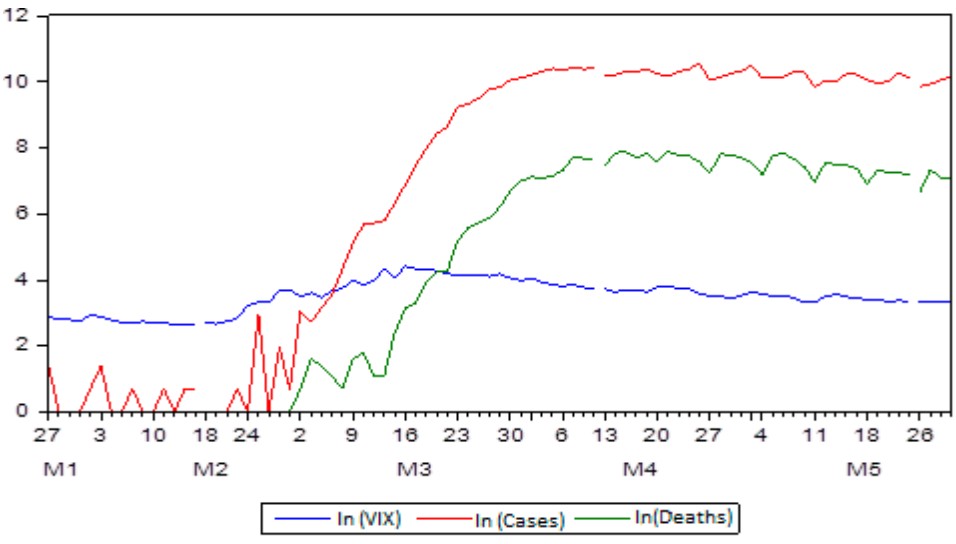

**Figure 1.** Natural logarithms of the VIX index and daily new case and death numbers in the USA.

It can be noted from Figure 1 that the VIX index increased with the increase in the number of the COVID-19 case and death numbers in the USA. Furthermore, the VIX fear index started to increase with the first pandemic case in the USA. Moreover, the VIX index made a significant leap even when there were no deaths. The VIX index reached the highest level before the pandemic contamination and death figures peaked. This could have been because the COVID-19 case numbers were not always accepted as fact by some people. As the pandemic cases and death numbers became more and more evident, the level of fear began to drop from the peak. The VIX index reached its peak by mid-March 2020, while the number of cases and deaths peaked at the end of March 2020. On the other hand, by

the third week of March, one could be witness that the stock markets studied here fell to their lowest levels.

To examine the relationship between the VIX index and daily new case and death numbers of the COVID-19 pandemic, we first needed to determine whether the time series was constant. The unit root properties of the time series used in the study were investigated using the unit root tests developed by Dickey and Fuller (1979) (augmented Dickey–Fuller (ADF)) and Phillips and Perron (1988) (PP). The unit root test results are given in Table 1.

**Table 1.** Unit root test results.

| Variables | Augmented Dickey–Fuller (ADF) Test | | | | Phillips–Perron (PP) Test | | | | Stationary Level |
|---|---|---|---|---|---|---|---|---|---|
| | Level | | Difference | | Level | | Difference | | |
| | Intercept | Trend and Intercept | Intercept | Trend and Intercept | Intercept | Trend and Intercept | Intercept | Trend and Intercept | |
| ln(VIX) | −1.43 | −1.02 | −10.81 *** | −11.06 *** | −1.45 | −1.01 | −10.66 *** | −10.89 *** | I(1) |
| ln(cases) | −2.51 | −2.24 | −7.15 *** | −18.24 *** | −1.05 | −1.19 | −16.87 *** | −17.38 *** | I(1) |
| ln(deaths) | −1.47 | −1.54 | −2.04 | −2.21 | −1.03 | −0.47 | −8.50 *** | −8.54 *** | I(1) |

*** Indicates statistical significance at the 1% level.

In the ADF (augmented Dickey–Fuller) and PP (Philips–Perron) tests, $H_1$ (the basic hypothesis) was established as the series had a unit root. That is, it was not stationary. Later after Table 1 was examined further, it was determined through the results of the ADF test that the level values of the VIX and COVID-19 case and death variables were not statistically significant and contained a unit root. Therefore, $H_1$ was rejected. A unit root was a stochastic trend in a time series, which was explained as a random walk with drift, and if the time series has a unit root as in this case, it means that there was an unpredictable systematic pattern and, therefore, one cannot regress the data since it had no meaning.

The Phillips–Perron test statistics also provided results that supported the ADF test statistics. It can therefore be concluded that the non-stationary variables in the level values did not have a unit root in their first differences, that is, their integration degrees were I(1) (Phillips and Perron 1988).

The fact that the series were integrated with the same degree does not mean that they always acted together in the long term. After determining that they were stationary in the first differences of the series, the existence of a long-term equilibrium relationship between the series was investigated according to the co-integration method developed by Johansen (1988) and Johansen and Juselius (1990). The Johansen co-integration test is based on vector autoregression model (VAR) analysis. The VAR model with a lagged k is shown as follows (Brooks 2008):

$$y_t = \beta_1 y_{t-1} + \beta_2 y_{t-2} + \cdots + \beta_k y_{t-k} + u_t \tag{1}$$

To use the Johansen and Juselius (1990) test, the VAR model must be converted to an error correction model (VECM), as follows:

$$\Delta y_t = \Pi y_{t-k} + \Gamma_1 \Delta y_{t-1} + \Gamma_2 \Delta y_{t-2} + \cdots + \Gamma_{k-1} \Delta y_{t-(k-1)} + u_t \tag{2}$$

$\Gamma$ and $\Pi$ represent coefficient matrices. Coefficient matrix $\Pi$ contains information about the long-term relationships. In the Johansen and Juselius co-integration method, the trace and maximum eigenvalue statistics are examined to reveal the existence of a co-integration relationship and the number of cointegrated vectors.

To perform a co-integration test, it is necessary to determine the appropriate lag length first. The appropriate lag length was determined by estimating an unrestricted VAR model with the variables used in the analysis. In determining the appropriate lag length, the LR (likelihood ratio), FPE (final prediction error), AIC (Akaike information criterion) and SC (Schwarz) and HQ (Hannan–Quinn) criteria were used. The result of the VAR lag length selection criteria is presented in Table 2.

**Table 2.** VAR lag length selection criteria.

| Lag | LR | FPE | AIC | SC | HQ |
|---|---|---|---|---|---|
| 0 | NA | 0.723416 | 8.189857 | 8.279836 | 8.225905 |
| 1 | 565.1489 | 0.000485 | 0.882386 | 1.242302 | 1.026580 |
| 2 | 50.95824 | 0.000301 | 0.402481 | 1.032334 * | 0.654819 * |
| 3 | 14.48562 | 0.000307 | 0.420392 | 1.320183 | 0.780876 |
| 4 | 14.82427 | 0.000309 | 0.423630 | 1.593358 | 0.892259 |
| 5 | 25.41032 | 0.000262 | 0.248140 | 1.687805 | 0.824914 |
| 6 | 14.88065 | 0.000259 * | 0.227977 * | 1.937579 | 0.912896 |
| 7 | 4.938314 | 0.000304 | 0.369188 | 2.348727 | 1.162252 |
| 8 | 17.07307 * | 0.000284 | 0.280869 | 2.530344 | 1.182077 |

* Indicates the appropriate lag length.

The appropriate lag length for the estimated VAR model was 6 according to the FPE and AIC criteria and 2 according to the SC and HQ criteria. Inverse roots of the AR characteristic polynomial were investigated regarding whether the estimated VAR model for 2 lag lengths contained a unit root. In Figure 2, we can see that all the inverse roots of the characteristic polynomial of the AR were located within the unit circle. The fact that the inverse roots were located in the unit circle shows that the predicted model displayed a stationary structure.

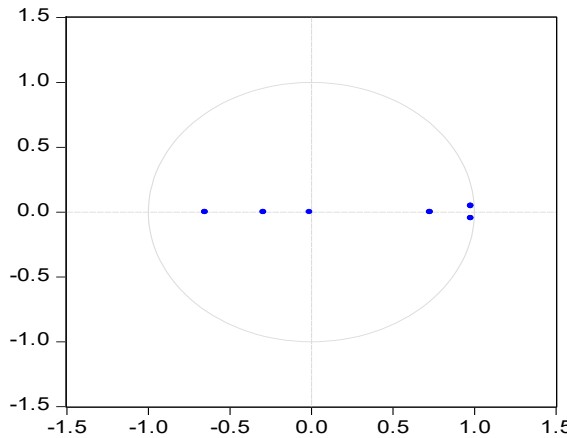

**Figure 2.** Inverse roots of the AR characteristic polynomial.

The Johansen co-integration test was used to determine the existence of a long-term relationship between the VIX index and daily new case and death numbers of the COVID-19 pandemic in the United States. The co-integration test results are given in Table 3.

**Table 3.** Co-integration test results.

| Hypothesis | Trace | 0.05 | Prob. ** | Max-Eigen | 0.05 | Prob. ** |
|---|---|---|---|---|---|---|
| | Statistic | Critical Value | | Statistic | Critical Value | |
| r = 0 * | 59.58050 | 35.19275 | 0.0000 | 43.58899 | 22.29962 | 0.0000 |
| r ≤ 1 | 15.99150 | 20.26184 | 0.1748 | 9.531600 | 15.89210 | 0.3790 |
| r ≤ 2 | 6.459904 | 9.164546 | 0.1581 | 6.459904 | 9.164546 | 0.1581 |

Trace test indicates one co-integrating equation(s) at the 0.05 level; * denotes rejection of the hypothesis at the 0.05 level; ** (MacKinnon et al. 1999) *p*-values.

According to the maximum eigenvalue and trace statistics obtained as a result of the Johansen co-integration test, the $H_1$ hypothesis was rejected. In other words, the hypothesis predicted that there was at least one co-integration vector accepted at the

1% significance level. According to these results, it is possible to state that a long-term equilibrium relationship was valid between the VIX index and the daily new case and death numbers of the COVID-19 pandemic in the United States during the analysed period.

After the long-term relationship between the VIX index and the case and death numbers was determined, the fully modified least-squares (FMOLS) estimator was used to estimate the long-term coefficients of each variable. While there was a co-integration relationship between the variables, there was a problem of correlation and endogeneity between the explanatory variables and the error terms. In this case, the variables lost their asymptotic properties (Berke 2012).

The FMOLS method developed by Phillips and Hansen (1990) took into account the autocorrelation and endogeneity problems arising from the co-integration relationship between the variables. FMOLS estimators are asymptotically deviated and have an asymptotically normal distribution (Phillips and Hansen 1990; Shahbaz 2009). The FMOLS model equation is as follows:

$$VIX = \alpha_1 + \alpha_{11}Case + \alpha_{12}Death + \varepsilon \qquad (3)$$

In the equation, $\varepsilon$ represents the error term of the model. At the end of the test, we checked and examined which variables were effective on the *VIX* index and the effect size of the variables affecting the *VIX* index. The results of the analyses are presented in Table 4.

**Table 4.** FMOLS test results.

| VIX | | |
|---|---|---|
| FMOLS | | |
| **Variables** | **Coefficient** | **t-Statistic** |
| C | 2.721101 | 38.52418 *** |
| Case | 0.325438 | 10.28894 *** |
| Death | −0.330127 | −8.205153 *** |
| $R^2$ = 0.753883 Adj. $R^2$ = 0.747952 | | |

*** Indicates statistical significance at the 1% level.

When the FMOLS test results in Table 4 were examined, it was seen that there was a statistically significant and long-run positive relationship between the cases and the VIX index. A 1% increase in cases affected the VIX index, which increased by 32.54%.

On the other hand, a statistically significant and negative relationship was found between the number of deaths and the VIX index in the long term at the 1% significance level. A 1% increase in the death variable affected the VIX index by a decrease of 33.01%.

The reason for this different result was that the VIX index was primarily affected by the increase in the COVID-19 case numbers and less by the death numbers. This may have been because the effect of the COVID-19 deaths was registered only after the case was diagnosed, i.e., there was a delay, and the deaths were expected as there was an increasing number of cases. Therefore, deaths had already been reflected in prices. This result complied with the expectation hypothesis and a common expression in the financial markets "Expectations are bought, facts are sold".

The high $R^2$ value (0.75), which was found for the diagnostic tests of the model, showed that 75% of the changes in the dependent variable could be explained by the changes in the independent variables. This was evidence of the suitability of the models. According to the results shown in Table 4, the $H_2$ hypothesis was accepted. This showed that bad news about COVID cases and deaths in the USA affected the VIX index.

### 4.2. The Effect of the VIX Index on the Major Stock Market Indexes during the Pandemic Period

The figures of the VIX, DJIA, DAX, CAC40, FTSE 100, SSEC, Nikkei 225 and MIB indexes are given in Appendix A.

From Appendix A, one can observe that there were decreases in major stock indices during the periods when the VIX index increased. This means that the risk perception that arose due to the COVID-19 cases increased and investor confidence in the markets decreased with investors moving away from the stock markets to avoid risk.

To examine the relationship between variables (VIX and stock indexes), it was necessary to determine whether the time series was stationary. Unit root test results are given in Table 5.

**Table 5.** Unit root test results.

| Variables | Augmented Dickey–Fuller (ADF) Test | | | | Phillips–Perron (PP) Test | | | | Stationary Level |
| --- | --- | --- | --- | --- | --- | --- | --- | --- | --- |
| | Level | | Difference | | Level | | Difference | | |
| | Intercept | Trend and Intercept | Intercept | Trend and Intercept | Intercept | Trend and Intercept | Intercept | Trend and Intercept | |
| ln(VIX) | 1.55 | −0.91 | −11.68 *** | −11.83 *** | −1.56 | −0.87 | −11.55 *** | −11.68 *** | I(1) |
| ln(DJI) | −1.25 | −0.77 | −14.54 *** | −14.58 *** | −1.42 | −1.52 | −13.88 *** | −13.93 *** | I(1) |
| ln(DAX) | −1.30 | −0.83 | −9.69 *** | −9.73 *** | −1.44 | −1.15 | −9.79 *** | −9.82 *** | I(1) |
| ln(CAC40) | −1.27 | −1.06 | −9.97 *** | −9.99 *** | −1.36 | −1.38 | −10.05 *** | −10.08 *** | I(1) |
| ln(FTSE100) | −1.36 | −1.07 | −10.15 *** | −10.18 *** | −1.35 | −1.13 | −10.15 *** | −10.18 *** | I(1) |
| ln(SSEC) | −2.21 | −2.31 | −9.83 *** | −9.82 *** | −2.26 | −2.44 | −9.83 *** | −9.82 *** | I(1) |
| ln(Nikkei225) | −1.22 | −0.57 | −8.13 *** | −8.23 *** | −1.37 | −0.91 | −8.17 *** | −8.23 *** | I(1) |
| ln(MIB) | −1.16 | −1.13 | −10.97 *** | −10.97 *** | −1.25 | −1.43 | −10.98 *** | −10.97 *** | I(1) |

*** Indicates statistical significance at the 1% level.

When Table 5 is examined, the ADF and PP unit root test results show that the series had a unit root. Thus, it was concluded that the null hypothesis that there was a unit root in the series was accepted and the level values of the series were not stable. When the first difference of the series was taken, the null hypothesis was rejected at the 1% significance level. Thus, the series became stationary at the first difference I(1).

After determining that the series were stationary in the first differences, the existence of a long-term equilibrium relationship between the VIX index and the major stock indexes was investigated using the co-integration method. To perform the co-integration test, it was necessary to estimate an unrestricted VAR model with the variables used in the model and determine the lag length of the model. To determine the co-integration relationship between the major stock indexes and the VIX index, binary VAR models were first created and appropriate lag lengths were determined. According to the SC criterion, the lag lengths were determined as 2 for the DJI–VIX index, 1 for the DAX–VIX index, 1 for the CAC40–VIX index, 1 for the FTSE100–VIX index, 1 for the SSEC–VIX index, 1 for the Nikkei225–VIX index and 1 for the MIB–VIX index (see Appendix B).

Inverse roots of the AR characteristic polynomial were investigated to determine whether the VAR models estimated by the lag lengths contained unit roots. We note that all of the inverse roots of the characteristic polynomial of the AR were located within the unit circle. The fact that the inverse roots were located in the unit circle shows that the predicted model displayed a stationary structure (see Appendix B).

The Johansen co-integration test was used to determine the existence of the long-term relationship between the VIX index and the major stock indexes. The co-integration test results with the determined lag lengths are given in Table 6.

**Table 6.** Co-integration test results.

| | | | | | | |
|---|---|---|---|---|---|---|
| **DJI–VIX** | | | | | | |
| **Hypothesis** | **Trace** | **0.05** | **Prob. \*\*** | **Max-Eigen** | **0.05** | **Prob. \*\*** |
| | **Statistic** | **Critical Value** | | **Statistic** | **Critical Value** | |
| r = 0 * | 23.81726 | 20.26184 | 0.0155 | 20.74610 | 15.89210 | 0.0079 |
| r ≤ 1 | 3.071160 | 9.164546 | 0.5675 | 3.071160 | 9.164546 | 0.5675 |
| **DAX–VIX** | | | | | | |
| **Hypothesis** | **Trace** | **0.05** | **Prob. \*\*** | **Max-Eigen** | **0.05** | **Prob. \*\*** |
| | **Statistic** | **Critical Value** | | **Statistic** | **Critical Value** | |
| r = 0 * | 21.68141 | 20.26184 | 0.0317 | 18.38857 | 15.89210 | 0.0199 |
| r ≤ 1 | 3.292837 | 9.164546 | 0.5274 | 3.292837 | 9.164546 | 0.5274 |
| **CAC40–VIX** | | | | | | |
| **Hypothesis** | **Trace** | **0.05** | **Prob. \*\*** | **Max-Eigen** | **0.05** | **Prob. \*\*** |
| | **Statistic** | **Critical Value** | | **Statistic** | **Critical Value** | |
| r = 0 | 15.35487 | 20.26184 | 0.2067 | 12.46625 | 15.89210 | 0.1605 |
| r ≤ 1 | 2.888624 | 9.164546 | 0.6017 | 2.888624 | 9.164546 | 0.6017 |
| **FTSE100–VIX** | | | | | | |
| **Hypothesis** | **Trace** | **0.05** | **Prob. \*\*** | **Max-Eigen** | **0.05** | **Prob. \*\*** |
| | **Statistic** | **Critical Value** | | **Statistic** | **Critical Value** | |
| r = 0 * | 17.87669 | 15.49471 | 0.0215 | 15.61454 | 14.26460 | 0.0304 |
| r ≤ 1 | 2.262146 | 3.841466 | 0.1326 | 2.262146 | 3.841466 | 0.1326 |
| **SSEC–VIX** | | | | | | |
| **Hypothesis** | **Trace** | **0.05** | **Prob. \*\*** | **Max-Eigen** | **0.05** | **Prob. \*\*** |
| | **Statistic** | **Critical Value** | | **Statistic** | **Critical Value** | |
| r = 0 * | 18.80945 | 15.49471 | 0.0152 | 16.89695 | 14.26460 | 0.0187 |
| r ≤ 1 | 1.912497 | 3.841466 | 0.1667 | 1.912497 | 3.841466 | 0.1667 |
| **Nikkei225–VIX** | | | | | | |
| **Hypothesis** | **Trace** | **0.05** | **Prob. \*\*** | **Max-Eigen** | **0.05** | **Prob. \*\*** |
| | **Statistic** | **Critical Value** | | **Statistic** | **Critical Value** | |
| r = 0 * | 23.77301 | 20.26184 | 0.0158 | 20.94368 | 15.89210 | 0.0073 |
| r ≤ 1 | 2.829334 | 9.164546 | 0.6131 | 2.829334 | 9.164546 | 0.6131 |
| **MIB–VIX** | | | | | | |
| **Hypothesis** | **Trace** | **0.05** | **Prob. \*\*** | **Max-Eigen** | **0.05** | **Prob. \*\*** |
| | **Statistic** | **Critical Value** | | **Statistic** | **Critical Value** | |
| r = 0 | 14.60842 | 20.26184 | 0.2497 | 10.91640 | 15.89210 | 0.2581 |
| r ≤ 1 | 3.692014 | 9.164546 | 0.4599 | 3.692014 | 9.164546 | 0.4599 |

Trace test indicates one co-integrating equation(s) at the 0.05 level; * denotes the rejection of the hypothesis at the 0.05 level; ** (MacKinnon et al. 1999) *p*-values.

From Table 6, we can determine from the maximum eigenvalue and trace statistics obtained as a result of the Johansen co-integration test for the VIX index and major stock indexes, with the exception for the CAC40 and the MIB indexes, that the $H_3$ hypothesis should be rejected. In other words, it was shown that the hypothesis that predicted that there was at least one co-integration vector was accepted at the 1% level of significance. According to these results and during the analysis period, it is possible to mention that there was a long-term equilibrium relationship between the major stock indexes and the VIX index, except in the case of the CAC40 and the MIB indexes.

After the long-term relationship between the major stock market indexes and the VIX index was determined, the FMOLS estimator was used to estimate the long-term coefficients of each variable. The FMOLS model equations were as follows:

$$DJI = \alpha_2 + \alpha_{21} VIX + \varepsilon \tag{4}$$

$$DAX = \alpha_3 + \alpha_{31} VIX + \varepsilon \tag{5}$$

$$FTSE100 = \alpha_4 + \alpha_{41} VIX + \varepsilon \tag{6}$$

$$SSEC = \alpha_5 + \alpha_{51} VIX + \varepsilon \tag{7}$$

$$Nikkei225 = \alpha_6 + \alpha_{61} VIX + \varepsilon \tag{8}$$

In the equations, $\varepsilon$ represents the error term of the model. With the *FMOLS* models, we aimed to determine which major stock indexes affected the *VIX* index and their effect size. The results of the analyses are presented in Table 7.

**Table 7.** FMOLS test results.

| DJIA | | |
|---|---|---|
| | FMOLS | |
| **Variables** | **Coefficient** | **t-Statistic** |
| C | 10.80244 | 275.6797 *** |
| VIX | −0.199304 | −17.33294 *** |
| | $R^2 = 0.877027$ Adj. $R^2 = 0.875797$ | |
| **DAX** | | |
| | FMOLS | |
| **Variables** | **Coefficient** | **t-Statistic** |
| C | 10.12313 | 222.0833 *** |
| VIX | −0.232615 | −17.36300 *** |
| | $R^2 = 0.886653$ Adj. $R^2 = 0.885497$ | |
| **FTSE100** | | |
| | FMOLS | |
| **Variables** | **Coefficient** | **t-Statistic** |
| C | 9.501245 | 199.9433 *** |
| VIX | −0.221475 | −15.87110 *** |
| | $R^2 = 0.872368$ Adj. $R^2 = 0.871066$ | |
| **SSEC** | | |
| | FMOLS | |
| **Variables** | **Coefficient** | **t-Statistic** |
| C | 8.119017 | 283.0218 *** |
| VIX | −0.044509 | −5.333397 *** |
| | $R^2 = 0.451951$ Adj. $R^2 = 0.445928$ | |
| **Nikkei225** | | |
| | FMOLS | |
| **Variables** | **Coefficient** | **t-Statistic** |
| C | 10.54448 | 356.5324 *** |
| VIX | −0.178228 | −20.55148 *** |
| | $R^2 = 0.888014$ Adj. $R^2 = 0.886797$ | |

*** Indicates statistical significance at the 1% level.

The results presented in Table 7 show that the $H_4$ hypothesis should be accepted and that as the VIX index rose, the prices of major world stock markets decreased. After examining the results of the FMOLS regression models in Table 7, it can be noted that the VIX index significantly affected the DJIA, DAX, FTSE100, SSEC and Nikkei225 indexes. A

1% increase in the VIX index caused the following index decreases: DJI by 19.93%, DAX by 23.26%, FTSE100 by 22.14%, SSEC by 4.45% and Nikkei225 by 17.82%. According to the FMOLS results, the weakest impact of the VIX index was on the Chinese SSEC index (4.4%, $R^2$ = 0.451951). This was because, in other countries, the COVID-19 cases continued at a certain rate, while in China, the reports during the analysed periods were that its COVID-19 cases had been reduced to a minimal level and daily social and economic activities had returned to a new normal. We did not present a result of FMOLS for the CAC40 and the MIB since there was no co-integration between these indexes and the VIX index.

## 5. Conclusions

In this study, we had two objectives: (i) that of understanding the impact of COVID-19 cases on the VIX fear index during the period studied and (ii) that of determining the impact of the VIX on major stock indexes, specifically the DJIA, FTSE100, DAX, CAC40, FT-SEMIB, SSEC, Nikkei225 and MID. To do this, we examined the co-integration relationships between variables and determined the effect levels using the FMOLS regression analysis.

One of the best measures of fear in the markets is the VIX. This is derived from the price of Standard and Poor (S&P 500) index options; it provides an objective measure of real-time sentiment and market stress (Ritholtz 2020). A number of authors (Gilboa and Schmeidler 1989; Epstein and Schneider 2003) have developed models that suggest that the uncertainty of outcomes is related to risk and has a negative impact on market prices. They assumed that when faced with uncertainty, investors take a risk-averse approach and base their expectations on worst-case scenarios (maxmin expected utility).

Williams (2009) and Bird and Yeung (2010) noted that both changes in the VIX and the level of VIX are a proxy for uncertainty and explain how the market responds to earnings information. They found evidence that at times when there is high uncertainty, the market sentiment plays a role in counteracting the resultant pessimism of that uncertainty.

Our results highlighted the co-integration between the VIX "fear" index and the COVID-19 cases. A 1% increase in the COVID-19 cases increased the "fear" index by 32.5%. Furthermore, we found co-integration between the VIX index and the major indexes, except for the CAC 40 and the MIB. Increases in the VIX index led to a fall in the major indexes. The largest effect was on the DAX (−23.26%) and the FTSE100 (−22.15%), while the lowest effect was on the SSEC (China) (−4.45%).

As noted by Li (2019), this is seen as an unusual trend in the stock market, as she highlighted that the VIX index typically trades inversely with other indexes and bad news. Moreover, she noted that this implies that something else or some noise was driving the markets.

Moreover, we determined that the COVID-19 cases had a larger impact on the VIX index than the COVID-19 deaths. The effect of the VIX index on the German and the British stock markets was larger than on the U.S. and the Chinese stock markets. This may have been due to the fact that the Federal Reserve (FED) in the United States can manage USD volume and apply policy. Regarding China, this may be explained by the fact that they had announced that the COVID-19 cases decreased before they peaked in other parts of the world. Furthermore, China has its regional market structure, which can be noted from other studies, such as that by Özen and Tetik (2019).

Our findings corroborate earlier findings by, for example, Fernandes (2020), Yilmazkuday (2020) and Cheng (2020). The former found that the British markets declined by more than 35%, while the U.S. stock market plummeted by over 25%. He noted that British stocks were affected by the COVID-19 cases in the USA and the VIX index because of its integration with the US economy. While the latter author highlighted that starting in March, the cases of COVID-19 were increasing rapidly in Europe and there were reports of community spreading and deaths in the United States. The S&P 500 had declined slightly, while the VIX index had risen to 33 from 14. Moreover, Yilmazkuday (2020), showed that a 1% increase in the U.S. cumulative daily COVID-19 cases resulted in around a 0.01%

cumulative reduction in the S&P 500 Index after one day and around a 0.03% reduction after one month, with the largest observations found during March 2020.

On the other hand and different from our findings, Albulescu (2020a) found that the death ratio positively influenced the VIX index and that this influencing effect was stronger outside China. Zeren and HIZARCI (2020) found no co-integration with the FTSE MIB, CAC40 or DAX30. Onali (2020) argued that the COVID-19 crisis did not have an impact on the U.S. stock market returns between 8 and 9 April 2020, with the VIX negatively affecting the stock market returns. However, the main reason for this may be due to the use of data collected from different areas, such as in the case of Onali (2020) in which data was collected mainly from China, and to the different periods under observation.

The findings of the study clearly showed the impact of the uncertainty and fear in the market, which resulted from the unprecedented COVID-19 pandemic outbreak and the daily new cases and deaths on the CBOE volatility index. Moreover, the study showed that there was a significant effect of the VIX index on the DJIA, DAX, FTSE100, SSEC and Nikkei225 indexes, thus showing the indirect effect of fear in the market. The findings are of special importance to portfolio and fund managers, as well as risk managers, underwriters and actuaries.

One must however understand that the findings of this study are limited to the period under study and to the markets that have been analysed. Therefore, although they can be somewhat generalisable to other similar or even smaller markets, further studies should be carried out on other markets and in different periods to determine whether the effects were similar or specific to the market in question. In addition, COVID-19 data was available 7 days a week, while the financial data reflected the working days of the week. However, we assumed that this limitation would not have much effect on our analysis since, in normal circumstances, other news is still available 24 hours a day, 7 days a week. Furthermore, we tried to eliminate the noise from spillover effects, such as pandemic fatigue and news on available vaccines and medicines, as much as possible by taking a period at an early stage of the outcome of the pandemic; as such, we assumed that there was no noise effect in our findings.

Moreover, above all, one must note that all models have various assumptions, which users such as risk managers, portfolio managers and policymakers need to be aware of; these findings, which are dependent on the model assumptions and the strength of other factors (considered herein as noise), might result in a distorted version of the social reality. However, models such as these are needed to build expectations and to understand the determinants of deviations from findings and results obtained, which help us develop forecasts and risk management decisions.

**Author Contributions:** Conceptualization, E.Ö., S.G. and L.Ö.; methodology, L.Ö.; validation, S.G., I.R. and E.Ö.; formal analysis, L.Ö., S.G. and E.Ö.; data curation, L.Ö., S.G. and E.Ö.; writing—original draft preparation, L.Ö., S.G. and E.Ö.; writing—review and editing, S.G. and I.R. All authors have read and agreed to the published version of the manuscript.

**Funding:** This research received no external funding.

**Data Availability Statement:** Data available at https://www.worldometers.info/coronavirus/#countries (accessed on 13 May 2021), https://www.investing.com/indices/ (accessed on 13 May 2021) and https://www.investing.com/indices/volatility-s-p-500 (accessed on 13 May 2021).

**Acknowledgments:** There was no administrative support, technical support, or donations in kind.

**Conflicts of Interest:** The authors declare no conflict of interest.

## Appendix A

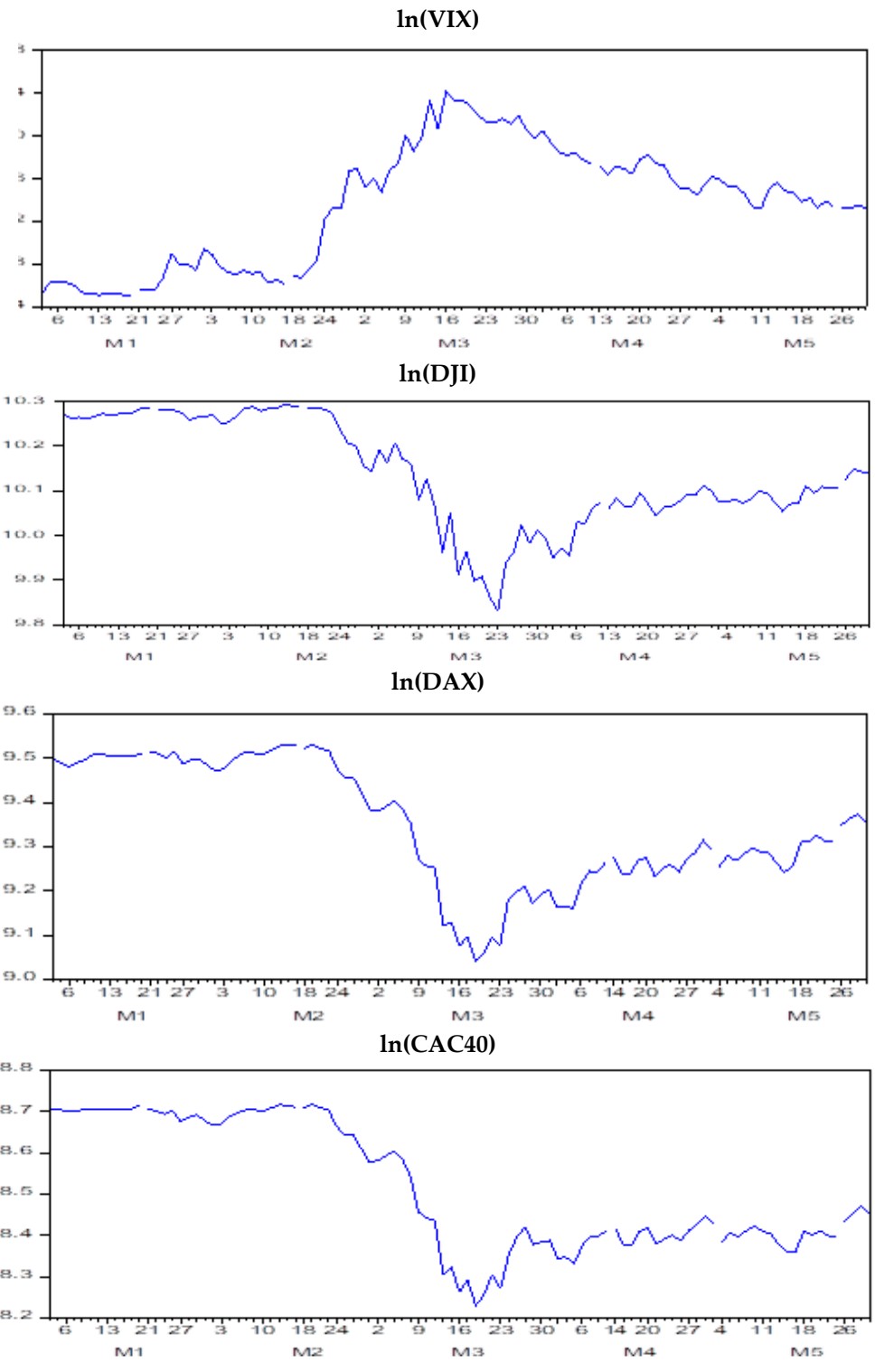

**Figure A1.** *Cont.*

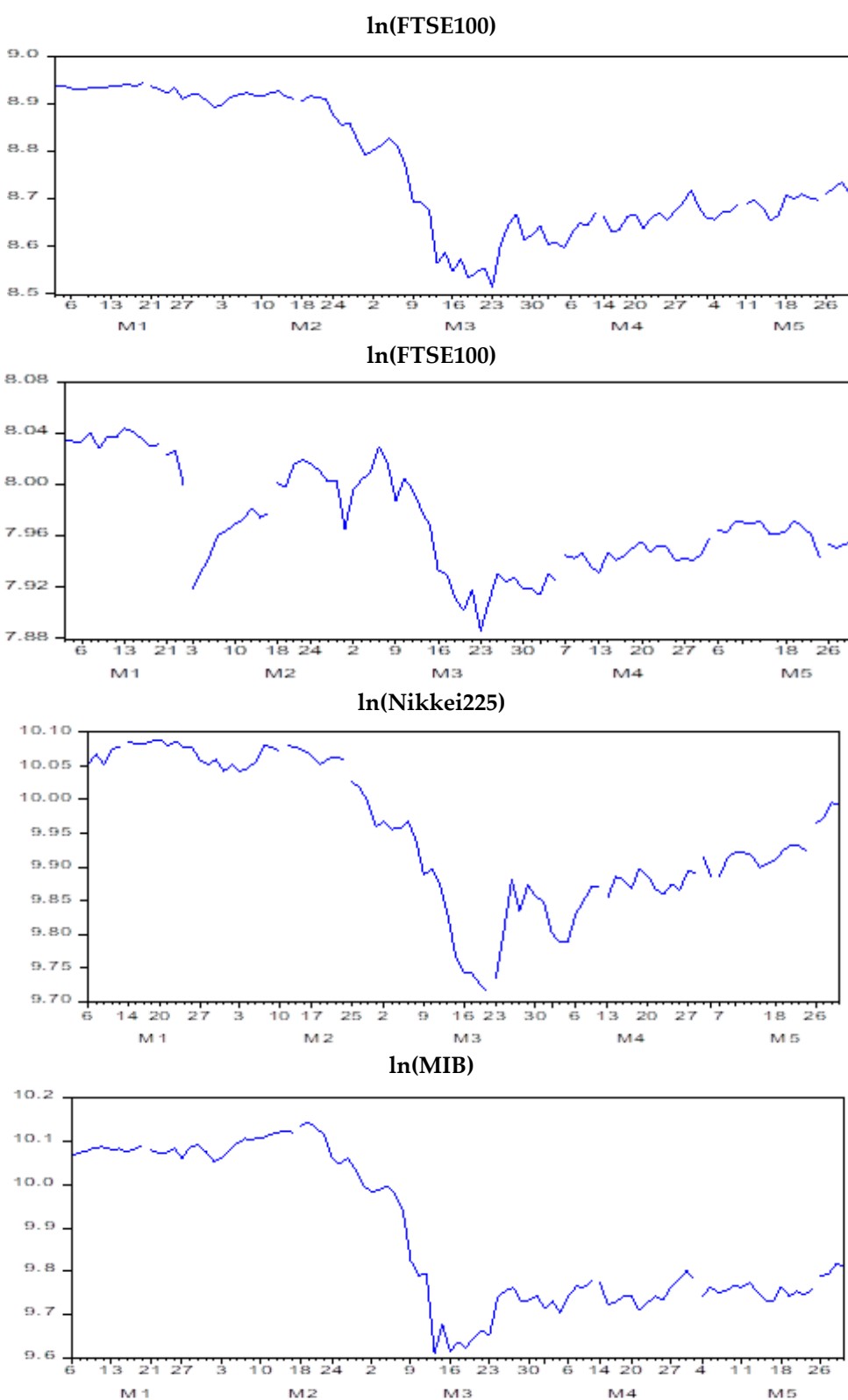

**Figure A1.** Series of VIX and major stock market indexes.

# Appendix B

**Table A1.** VAR lag length selection criteria.

| | | DJI–VIX | | | | | DAX–VIX | | | |
|---|---|---|---|---|---|---|---|---|---|---|
| Lag | LR | FPE | AIC | SC | HQ | LR | FPE | AIC | SC | HQ |
| 0 | NA | 0.0005 | −1.857 | −1.803 | −1.835 | NA | 0.0006 | −1.617 | −1.563 | −1.595 |
| 1 | 409.054 | $6.82 \times 10^{-6}$ | −6.219 | −6.057 | −6.154 | 423.469 | $6.71 \times 10^{-6}$ | −6.236 | −6.073 * | −6.170 * |
| 2 | 17.637 * | $6.10 \times 10^{-6}$ * | −6.331 * | −6.062 * | −6.222 * | 9.894 | $6.53 \times 10^{-6}$ * | −6.263 * | −5.990 | −6.153 |
| 3 | 4.604 | $6.30 \times 10^{-6}$ | −6.299 | −5.922 | −6.147 | 5.065 | $6.71 \times 10^{-6}$ | −6.236 | −5.854 | −6.082 |
| 4 | 4.948 | $6.48 \times 10^{-6}$ | −6.272 | −5.788 | −6.076 | 1.768 | $7.17 \times 10^{-6}$ | −6.171 | −5.680 | −5.973 |
| 5 | 6.426 | $6.53 \times 10^{-6}$ | −6.264 | −5.673 | −6.025 | 4.009 | $7.45 \times 10^{-6}$ | −6.134 | −5.534 | −5.892 |
| 6 | 1.119 | $7.02 \times 10^{-6}$ | −6.194 | −5.495 | −5.911 | 3.098 | $7.82 \times 10^{-6}$ | −6.086 | −5.378 | −5.800 |
| 7 | 7.432 | $6.97 \times 10^{-6}$ | −6.202 | −5.396 | −5.877 | 13.231 * | $7.21 \times 10^{-6}$ | −6.170 | −5.353 | −5.840 |
| 8 | 6.418 | $7.00 \times 10^{-6}$ | −6.201 | −5.286 | −5.831 | 2.932 | $7.58 \times 10^{-6}$ | −6.122 | −5.197 | −5.749 |

| | | CAC40–VIX | | | | | FTSE100–VIX | | | |
|---|---|---|---|---|---|---|---|---|---|---|
| Lag | LR | FPE | AIC | SC | HQ | LR | FPE | AIC | SC | HQ |
| 0 | NA | 0.001450 | −0.860 | −0.806 | −0.838 | NA | 0.000725 | −1.553 | −1.499 | −1.531 |
| 1 | 485.286 | $7.19 \times 10^{-6}$ | −6.166 | −6.003 * | −6.100 * | 442.02 | $5.82 \times 10^{-6}$ | −6.378 | −6.215 * | −6.312 * |
| 2 | 9.335 | $7.05 \times 10^{-6}$ * | −6.186 * | −5.914 | −6.076 | 8.499 | $5.76 \times 10^{-6}$ * | −6.389 * | −6.117 | −6.279 |
| 3 | 4.692 | $7.28 \times 10^{-6}$ | −6.155 | −5.774 | −6.001 | 4.723 | $5.94 \times 10^{-6}$ | −6.358 | −5.977 | −6.204 |
| 4 | 1.781 | $7.77 \times 10^{-6}$ | −6.090 | −5.600 | −5.892 | 2.267 | $6.31 \times 10^{-6}$ | −6.299 | −5.809 | −6.101 |
| 5 | 3.544 | $8.12 \times 10^{-6}$ | −6.047 | −5.448 | −5.805 | 1.701 | $6.74 \times 10^{-6}$ | −6.233 | −5.634 | −5.992 |
| 6 | 4.018 | $8.43 \times 10^{-6}$ | −6.011 | −5.303 | −5.726 | 2.550 | $7.12 \times 10^{-6}$ | −6.179 | −5.471 | −5.893 |
| 7 | 12.204 * | $7.87 \times 10^{-6}$ | −6.082 | −5.265 | −5.752 | 15.302 * | $6.39 \times 10^{-6}$ | −6.290 | −5.473 | −5.960 |
| 8 | 3.062 | $8.26 \times 10^{-6}$ | −6.036 | −5.110 | −5.662 | 2.843 | $6.73 \times 10^{-6}$ | −6.241 | −5.315 | −5.867 |

| | | SSEC–VIX | | | | | Nikkei225–VIX | | | |
|---|---|---|---|---|---|---|---|---|---|---|
| Lag | LR | FPE | AIC | SC | HQ | LR | FPE | AIC | SC | HQ |
| 0 | NA | 0.000237 | −2.670 | −2.613 | −2.647 | NA | 0.000387 | −2.182 | −2.125 | −2.159 |
| 1 | 378.70 * | $2.72 \times 10^{-6}$ | −7.140 | −6.968 * | −7.071 * | 356.73 * | $6.07 \times 10^{-6}$ | −6.337 | −6.166 * | −6.268 * |
| 2 | 9.253 | $2.66 \times 10^{-6}$ * | −7.161 * | −6.875 | −7.046 | 9.472 | $5.93 \times 10^{-6}$ * | −6.360 * | −6.077 | −6.246 |
| 3 | 1.132 | $2.88 \times 10^{-6}$ | −7.082 | −6.683 | −6.921 | 3.392 | $6.23 \times 10^{-6}$ | −6.311 | −5.914 | −6.151 |
| 4 | 3.812 | $3.01 \times 10^{-6}$ | −7.039 | −6.525 | −6.832 | 2.839 | $6.59 \times 10^{-6}$ | −6.255 | −5.745 | −6.050 |
| 5 | 1.283 | $3.25 \times 10^{-6}$ | −6.963 | −6.335 | −6.710 | 6.848 | $6.61 \times 10^{-6}$ | −6.253 | −5.630 | −6.002 |
| 6 | 2.744 | $3.44 \times 10{-6}$ | −6.907 | −6.165 | −6.609 | 2.599 | $7.01 \times 10^{-6}$ | −6.196 | −5.459 | −5.900 |
| 7 | 5.993 | $3.48 \times 10^{-6}$ | −6.899 | −6.042 | −6.554 | 4.440 | $7.24 \times 10^{-6}$ | −6.166 | −5.316 | −5.824 |
| 8 | 0.606 | $3.80 \times 10^{-6}$ | −6.814 | −5.844 | −6.424 | 8.771 | $7.03 \times 10^{-6}$ | −6.199 | −5.236 | −5.811 |

| | | MIB–VIX | | | |
|---|---|---|---|---|---|
| Lag | LR | FPE | AIC | SC | HQ |
| 0 | NA | 0.002137 | −0.472 | −0.417 | −0.450 |
| 1 | 489.63 | $8.95 \times 10^{-6}$ | −5.948 | −5.783 * | −5.881 |
| 2 | 12.488 | $8.45 \times 10^{-6}$ | −6.005 | −5.730 | −5.894 * |
| 3 | 8.284 | $8.36 \times 10^{-6}$ * | −6.016 * | −5.630 | −5.860 |
| 4 | 1.979 | $8.92 \times 10^{-6}$ | −5.952 | −5.456 | −5.752 |
| 5 | 2.507 | $9.45 \times 10^{-6}$ | −5.896 | −5.289 | −5.651 |
| 6 | 2.008 | $1.01 \times 10^{-5}$ | −5.834 | −5.116 | −5.544 |
| 7 | 11.415 * | $9.48 \times 10^{-6}$ | −5.896 | −5.068 | −5.562 |
| 8 | 2.473 | $1.00 \times 10^{-5}$ | −5.842 | −4.903 | −5.463 |

* Indicates the appropriate lag length.

## Appendix C

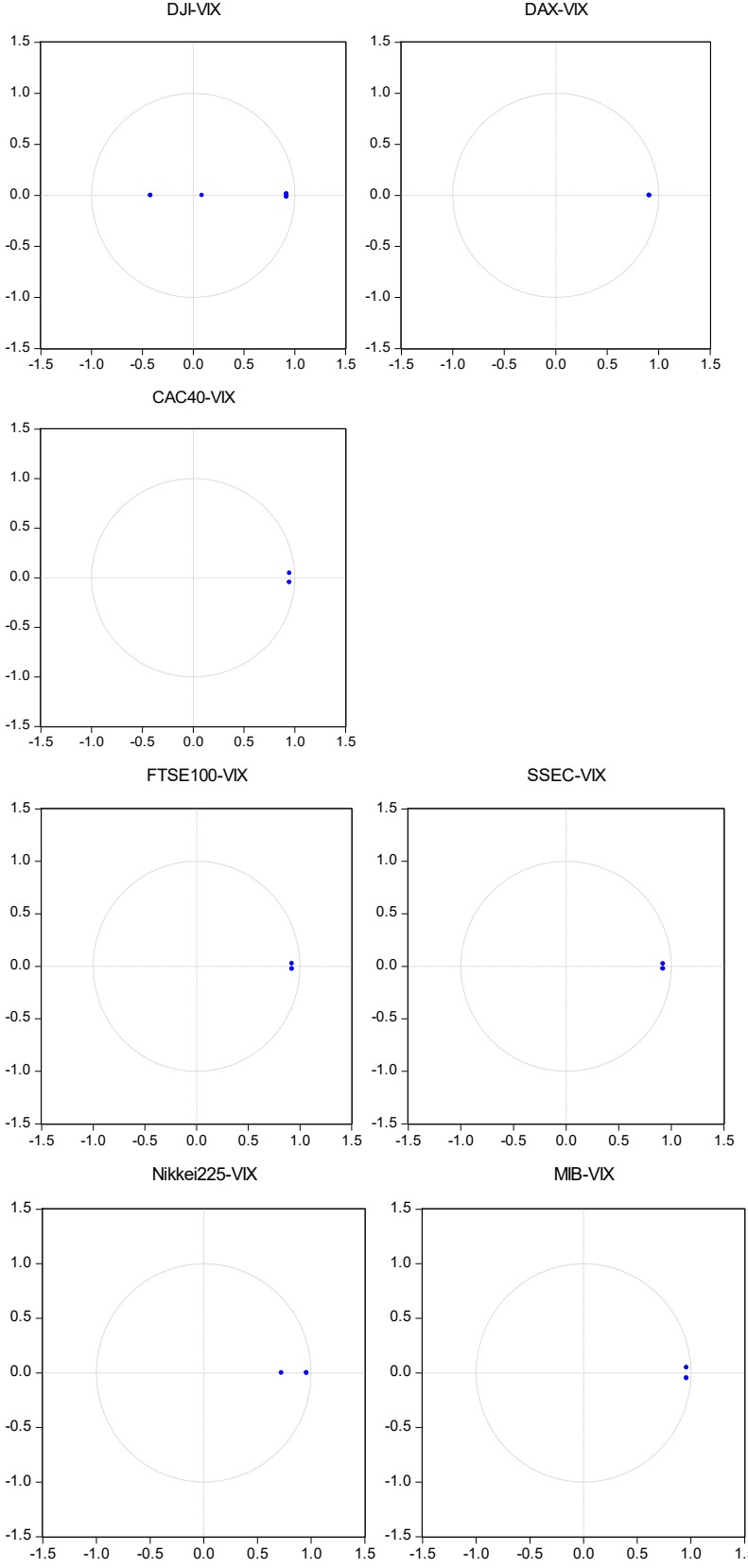

**Figure A2.** Inverse roots of the AR characteristic polynomials.

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
