# Peer review of "The Interactions between COVID-19 Cases in the USA, the VIX Index and Major Stock Markets"

_ijfs, doi:10.3390/ijfs9020026_

Round 1

Reviewer 1 Report

I don't understand why you started with instructions about publishing. Define the introduction in the introduction section and I suggest you remove the first paragraph

In your research you have discovered some similarity between the decree of the state of emergency and the volatility of the markets considering that both have a reference period of 30 days.

In order for your scientific research to confer applicability and comparability with the sociological situations described by the COVID 19 pandemic, please develop the Unit Root Test Results interpretation.

Often the mathematical models proposed by the authors are distorted by the social reality, I will ask you not to lose sight of the social character of the economic sciences and to link the mathematical approach to the social reality in America for the analyzed period.

Please as far as you can expand the conclusions section especially with reference to your research

Author Response

Dear Reviewer

Thank you for your kind comments and suggestions which we have taken on board and addressed to the best of our knowledge. These suggestions have helped to clarify some anomalies and made our paper much stronger. We have also proofread the paper to ensure that there are no spelling mistakes and the English grammar use is correct and appropriate.

For ease of the reader, we have herein added your comments and suggestions before our answers noted as 'S' for suggestion and 'A' for Answer.

S1: don't understand why you started with instructions about publishing. Define the introduction in the introduction section and I suggest you remove the first paragraph 

A1: Thank you for this comment you are correct this was an error on our part and the paragraph has been removed.

S2: In your research, you have discovered some similarities between the decree of the state of emergency and the volatility of the markets considering that both have a reference period of 30 days. In order for your scientific research to confer applicability and comparability with the sociological situations described by the COVID 19 pandemic, please develop the Unit Root Test Results interpretation.

A2:Thank you for this suggestion which has been taken on board. We have expanded clarifications on Unit root tests and their significance and interpretation.

S3:Often the mathematical models proposed by the authors are distorted by the social reality, I will ask you not to lose sight of the social character of the economic sciences and to link the mathematical approach to the social reality in America for the analyzed period. Please as far as you can expand the conclusions section especially with reference to your research

A3: Thank you for this suggestion which we have taken on board. This has made our paper much stronger. We understand that models have various assumptions which users such as risk managers, portfolio managers and policymakers need to be aware of, since as you rightly highlight can show a distorted version of the social reality. We have seen this clearly in cases such as LTCM which nearly brought down the whole US market. We have added some references and sentences in the conclusion to explain and discuss social reality and model results can defer.

All the changes have been track changed in our MS word Document for ease of reference.

Reviewer 2 Report

Dear authors,

I really appreciate your work and want to congratulate you for the research conducted, considering that the subject on COVID-19 is very discussed in our days.

I have a few minor recommendations for you: 

Keywords:-       it is not recommended to use the same words as in the title-       also the abbreviation is not necessary

Introduction -       line 22-30: I think this paragraph should not be part of the article, I think it was mistakenly forgotten here

Aim and Data - you should explain the choice of time period, 27 January 2020 to 29 May 2020 - you said it covered the period of the first wave - how do you know that the first wave is until May 29?

- I consider that the hypotheses should be numbered H1, H2, H3 and H4 for a better subsequent understanding of the results obtained -line 245, line 324…

Methodology and empirical results - in my opinion, the Figure 3 could be attached as appendix to the article (Similar to Appendix B).

Conclusions -        In the conclusions, the methodology used, the future research directions and limitations should be briefly presented

Author Response

Thank you for your kind comments and suggestions which we have taken on board and addressed to the best of our knowledge. These suggestions have helped to clarify some anomalies and made our paper much stronger. We have also proofread the paper to ensure that there are no spelling mistakes and the English grammar use is correct and appropriate.

For ease of the reader, we have herein added your comments and suggestions before our answers noted as 'S' for suggestion and 'A' for Answer.

S1:I really appreciate your work and want to congratulate you on the research conducted, considering that the subject on COVID-19 is very discussed in our days.

A1: Thank you for your nice words

S2:  Keywords:-       it is not recommended to use the same words as in the title-       also the abbreviation is not necessary

A2: Thank you for this suggestion which has been taken on board. The abbreviation has been removed and the key words changed as suggested

S3: Introduction -       line 22-30: I think this paragraph should not be part of the article, I think it was mistakenly forgotten here

A3: Thank you for this comment you are correct this was an error on our part and the paragraph has been removed.

S4: Aim and Data - you should explain the choice of the time period, 27 January 2020 to 29 May 2020 - you said it covered the period of the first wave - how do you know that the first wave is until May 29?

S5: Thank you for this comment, which we agree to. We have removed the words ‘first wave’ and elaborate further on the period chosen and the reason for this.

S5:  I consider that the hypotheses should be numbered H1, H2, H3 and H4 for a better subsequent understanding of the results obtained -line 245, line 324…

A5: Thank you for this comment, which we agree to and have arranged the hypotheses as suggested.

S6: Methodology and empirical results - in my opinion, Figure 3 could be attached as an appendix to the article (Similar to Appendix B).

A6: Thank you for this comment, which we agree to and put Figure 3 as Appendix A, Others B and C

S7: Conclusions -        In the conclusions, the methodology used, the future research directions and limitations should be briefly presented

A7: Thank you for this comment, which we agree to  and have revised the conclusion as per suggestions.

Also, all the changes have been track changed in our MS word Document for ease of reference.

Round 2

Reviewer 1 Report

The authors proved the entry of the observations in the new version of the article
I have no additional comments

Author Response

Dear Reviewer

Thank you for providing us with the opportunity to review our article in line with the very effective comments and suggestions of the peer reviewers, which has made our article much more relevant and stronger.

All comments and suggestions have been addressed and a total proofread has been carried out by native English speakers (Track changes were included).

We look forward to being published in the journal

Thank you

Kind Regards

The Authors
